# The Association between Socio-Demographics and Mental Distress Following COVID-19 Vaccination—Mediation of Vaccine Hesitancy

**DOI:** 10.3390/vaccines10101697

**Published:** 2022-10-11

**Authors:** Xiaoying Zhang, Junwei Shen, Ming Li, Yijian Shi, Qing Wang, Fazhan Chen, Hongyun Qin, Xudong Zhao

**Affiliations:** 1College of Public Health, Shanghai University of Medicine and Health Sciences, Shanghai 201318, China; 2Clinical Research Center for Mental Disorders, Shanghai Pudong New Area Mental Health Center, School of Medicine, Tongji University, Shanghai 200124, China; 3Department of Health Sciences, Towson University, Towson, MD 21252, USA; 4School of Education, Shanghai Normal University, Shanghai 201418, China; 5Department of Individual, Family, and Community Education, University of New Mexico, Albuquerque, NM 87131, USA

**Keywords:** COVID-19, vaccine hesitancy, anxiety, depressive symptoms, mediation analysis

## Abstract

The COVID-19 vaccine has been administered to over 200 countries and regions. With the unprecedented vaccination scale and speed, vaccination correlated mental health issues should be paid precise attention to. This study aims to assess the association between socio-demographic factors and mental health following vaccination and to analyze the mediation effect of vaccine hesitancy. This study recruited 2112 individuals who took two doses of the COVID-19 vaccine in Shanghai. Structural equation modeling was performed to assess factors associated with anxiety and depression of the vaccinated individuals and the underlying mechanism. The results yielded that vaccine hesitancy partially mediated/suppressed the effect from gender and employment status to anxiety/depression and fully mediated the effects from education to anxiety/depression. This study advanced the understanding of mental health disparity among different socio-demographic groups after vaccination and the impact of vaccine hesitancy on the vaccinated population’s mental health. The finding offered insights into the possible mental vulnerability of people holding a hesitant attitude before vaccination and suggested that vaccine hesitancy played a crucial role in people’s mental health after vaccination. Health promotion programs can target vaccine hesitancy to prevent unfavorable mental health consequences among specific populations.

## 1. Introduction

The coronavirus disease (COVID-19) outbreak has caused profound disruptions to the world economy and the loss of countless lives. Since its outbreak, two and half years have passed, but the pandemic still rages. The COVID-19 vaccine, emerging as an effective strategy to end the pandemic, has been rolled out in over 200 countries and regions. Till 1 September 2022, 67.7% of the world population has received at least one dose of the COVID-19 vaccine [1]. China launched the vaccine rollout in December 2020. As of 7 September 2022, 89.11% of the Chinese population has been fully vaccinated [1]. With such an immunization campaign worldwide, mental health following the vaccine campaign is a critical issue to consider.

However, the previous literature rarely examined mental health and the related factors among the vaccinated population. Previous studies indicated that mental health is associated with socio-demographic factors such as gender, education level, and employment status [2,3,4,5,6]. For instance, females were reported to relate to a higher level of anxiety and depression than males [7,8,9]. Unemployment status was also reported to correlate to worse mental health [2,3,4,5,6]. These associations might change or become more complex among the vaccinated population. It is interesting to explore the underlying association between socio-demographic factors and mental distress after vaccination, especially the effects of some vaccine-related factors.

Vaccine hesitancy, defined as “behavioral delay in acceptance or refusal of vaccines despite availability of vaccine services” [10], might be an inevitable contributing factor to consider for mental distress after vaccination. Because of the rapid development and novelty of the COVID-19 vaccines, vaccine hesitancy has become a common issue in China and other countries [11,12,13,14]. Vaccine hesitancy lies in the middle of a spectrum that ranges from full vaccination acceptance to vaccination refusal [10]. As stated in the previous literature, expressing reluctance to vaccination does not equate to not receiving the vaccine [15]. As the vaccine rollout has proceeded quickly in China, some people might hold a hesitant attitude toward the vaccine but still choose to be vaccinated. Vaccine hesitancy constitutes a type of uncertainty stress, which is defined as anxiety in facing ambiguous situations and environments. The previous literature has demonstrated that uncertainty was associated with various psychological disorders, including depression, anxiety, and psychiatric symptoms [16,17]. However, few studies have explored whether the uncertainty before vaccination influences mental health after vaccination [18,19]. It is interesting to investigate if vaccine hesitancy is associated with a higher level of anxiety or depression after vaccination.

As suggested by the theory of planned behavior, health beliefs/attitudes are the most proximal factors for health behavior/outcomes, and personal traits are the distal factors [20,21,22]. In this study, hesitancy over COVID-19 vaccination is a healthy attitude, which is a proximal factor for health. In addition, gender is a trait, and socio-economic status is among the social determinants of health, both of which are more dismal factors of health. Thus, health hesitancy may suppress/mediate between socio-demographic characteristics and mental health outcomes. For instance, females, who are reportedly more hesitant in making vaccination decisions [23,24,25,26], might develop a high level of anxiety/depression. Similarly, highly educated and employed, who are more uncertain [26,27,28], might also develop more significant anxiety/depressive symptoms. Mediation analysis effectively takes multiple factors into account and, most importantly, tests how the mediator intervenes in the pathway from socio-demographic characteristics to mental distress among the vaccinated population [29].

Given the background, the present study investigated the association between socio-demographics/vaccine hesitancy and subsequent mental health after vaccination. Furthermore, the mechanism between socio-demographic characteristics and mental health was explored. It was hypothesized that in addition to the direct effect of socio-demographic factors on mental health, there are three possible two-step indirect paths: (a) vaccine hesitancy would mediate between gender and anxiety/depression (i.e., female→higher hesitancy level→higher anxiety/depression level). (b) Vaccine hesitancy would mediate between education and mental distress (i.e., higher education level→higher hesitancy level→higher anxiety/depression level). (c) Vaccine hesitancy would mediate/suppress between employment status and mental status (i.e., employed/unemployed→higher hesitancy level→higher anxiety/depression level) (Figure 1).

## 2. Materials and Methods

### 2.1. Study Design and Setting

This retrospective study intended to assess factors associated with the mental health of people a period after taking up the first vaccine shot. The study was conducted in Shanghai, one of the largest metropolitans in China. Participants were recruited from multiple vaccination sites, including one university, one urban community health center, one suburban community health center, and one temporary vaccination site, from 27 April to 10 July 2021. The study was conducted in the observation room, and the target population was those who had just taken their second vaccine shot. The second-dose takers were chosen to ensure a certain period after the first dose, as there is a 3–8 weeks interval between the first and second dose. Eligible criteria required individuals to be (1) 18 years or older; (2) had just taken the second dose of the COVID-19 vaccine. People not meeting the criteria were excluded from the study. 

### 2.2. Data Collection Process

As mentioned earlier, the investigation was conducted in the observation room at each vaccination site with the permission of the organizers (e.g., the head of community health centers). The research team made two roll-up banners and flyers containing the study’s information and the questionnaire’s QR code. The research team approached each person in the waiting room and asked whether they had just taken the first or the second vaccine shot. People who had just taken their first shot were excluded. We further asked people who just took their second shot if they would like to participate in our study. Those who agreed to participate can scan the QR code by their mobile phone to fill in the questionnaire. Participants’ consent was recorded electronically by clicking the box before they answered the questions. Because we did not have the total number of people we approached, we could not calculate the refusal rate. As set in Wen Juan Xing, each mobile phone can only submit the questionnaire once to avoid repetitive submissions. 

If respondents were unfamiliar with smartphones, investigators assisted the answering process with permission from the respondents. Each vaccination site was staffed by a sufficient number of trained interviewers to ensure that each person in need could be assisted. Incentives such as alcohol sanitizing wipes were distributed to those who completed the questionnaire as compensation for their time. The first and second authors of the study were also presented at the vaccination site for each investigation. The investigation lasted for about three months, and 20 investigation events were conducted. In total, 2164 individuals finished the questionnaire. The final sample size for this study was 2112 after deleting incomplete surveys and those with unreliable responses (answer time < 120 s).

### 2.3. Questionnaire Development and Measures

A self-administered questionnaire was designed upon a thorough examination of the previous literature [11,18,30] by the research team composed of psychiatrists, health educators, epidemiologists, and social and behavioral sciences researchers. The questionnaire was designed on the Wen Juan Xing platform (Changsha Ranxing Information Technology Co., Ltd., Hunan, China), the largest questionnaire survey platform in China.

The contents of the questionnaire included (1) socio-demographic characteristics, including age, gender, marital status, education level, employment status, and areas of living (rural or urban); (2) level of hesitancy before the first shot of the vaccine; (3) depressive and anxiety symptoms at least three weeks after taking the first COVID-19 vaccine shot (measured after they took the second shot). All questions were closed-ended, with tick boxes provided for responses. 

The level of hesitancy was assessed on a five-point Likert scale using the following question, “were you hesitant before receiving the first COVID-19 vaccine shot?” [1 = not at all, 5 = very hesitant]. It was developed on previous literature [11,19] and through discussion with mental health professionals and researchers (XZ and FC) in the research team. 

Depressive symptoms were assessed using the 9-item Patient Health Questionnaire (PHQ-9), a self-reported questionnaire to evaluate depression by assessing one’s state over the previous two weeks [31]. The PHQ-9 is a 9-item questionnaire with a 4-point response scale ranging from 0 (never) to 3 (almost every day), and the total theoretical score ranges from 0 to 27. PHQ-9 was used as a continuous variable in this study. We used 9/10 as a cutoff to calculate the rate of depression among our sample. The validity and reliability of the Chinese version of PHQ-9 have been examined in the general Chinese population [32]. The Cronbach’s alpha for PHQ-9 was 0.92 in our sample, indicating a high internal consistency. 

Anxiety was assessed using the seven-item Generalized Anxiety Disorder Assessment (GAD-7) scale [33]. The GAD-7 is a 7-item questionnaire with a 4-point response scale ranging from 0 (never) to 3 (almost every day), and the total theoretical score ranges from 0 to 21. The summed score of the seven questions was then used to measure the anxiety level. We used 9/10 as a cutoff to calculate the rate of anxiety among our sample. GAD-7 was introduced in China and tested for good validity and reliability among the Chinese population [34]. The Cronbach’s alpha for GAD-7 was 0.93 in our sample. 

### 2.4. Ethics Statement

This study was approved by the ethics review committee of Shanghai Pudong New Area Mental Health Center Affiliated to Tongji University (No: PDJWLL2021027). The information sheet was provided, and consent was recorded electronically by clicking a box before answering the questions.

### 2.5. Statistical Analysis

Stata 16.0 (Stata Corp, College Station, TX, USA) was used for data management and analysis. Data cleaning was performed before the analysis. Descriptive statistics were performed to describe the participants’ socio-demographic characteristics, vaccine hesitancy, depressive symptoms, and anxiety status. Spearman’s correlation was tested between demographics and vaccine hesitancy, anxiety, and depressive scores. Only those demographics with significant correlations with vaccine hesitancy, anxiety, or depressive scores were included in the path analysis model as covariates. 

Two SEM models were constructed to check the model fit, the factors correlated to vaccine hesitancy and mental distress, and the mediation effect of vaccine hesitancy. Age, gender, employment status, marital status, and education level were controlled in each model for the mental health indicators and vaccine hesitancy. In this study, *p* values < 0.05 were considered statistically significant. The maximum likelihood method was used to fit the initial model. Model fit was assessed based on three fit indices: the root mean square error of approximation (RMSEA), comparative fit index (CFI), and standardized root mean residual (SRMR). A RMSEA < 0.08, a CFI > 0.90, and a SRMR < 0.06, was adopted as the cut-off point for an adequate model fit [35]. Path coefficients were used to examine the paths of demographics on anxiety and depressive level (including direct and indirect effects). Sobel tests were used to analyze the mediating effect of vaccine hesitancy. 

## 3. Results

### 3.1. Characteristics of the Study Sample

A total of 2112 adults who met the inclusion criteria were included in the analysis. As presented in Table 1, the majority of the study participants were young adults (63.26% ≤ 25 years old) or middle-aged (26.70% between 26 to 40 years), unmarried (78.08%), students (64.65%) or employed (31.30%), with a high school degree or above (89.25%) and living in urban areas (91.05%). Over half of the participants are female (53.60%). About one-third (33.29%) of the participants reported “somewhat hesitant” or “very hesitant” before receiving their first vaccine shot (Table 1).

The mean score of PHQ-9 was 1.95; 84.52% of the participants scored 0–4, indicating no depressive symptoms; 11.74% scored 5–9, indicating mild depression, and 3.74% of the participants had a depressive score higher than 9, which indicated moderate or major depression [31]. The mean score of the GAD-7 was 1.51; 86.93% of the participants scored 0–4, indicating no generalized anxiety, 11.74% scored 5–9, indicating mild anxiety, and 2.79% of the participants had an anxiety score higher than 9, indicating a moderate or major generalized anxiety disorder [33] (Table 1).

### 3.2. Correlations among Demographics, Vaccine Hesitancy, and Mental Health

#### 3.2.1. Correlation Analysis

Spearman’s correlation among the socio-demographic variables, vaccine hesitancy, and anxiety/depression are shown in Table 2. As expected, all socio-demographic variables were significantly associated with vaccine hesitancy and anxiety/depression (*p* < 0.05). 

We used structural equation modeling methods to test the proposed models for the mediation effect of vaccine hesitancy between demographic variables and mental distress. The goodness of fit of the two models was desirable. For depressive symptoms model, RMSEA = 0.000; CFI = 1.000; SRMR = 0.000; and for anxiety model, RMSEA = 0.000; CFI = 1.000; SRMR = 0.000. 

#### 3.2.2. Factors of Depression and the Mechanism

Adjusted for background factors, gender (males as reference) (β = 0.06, *p* < 0.01) and employment status (employed as reference) (β = 0.12, *p* < 0.001) was significantly associated with depressive symptoms. In addition, vaccine hesitancy was positively associated with depressive symptoms (β = 0.17, *p* < 0.001) (Figure 2). 

Figure 2 and Table 3 present the mechanism underlying the correlation between gender/education/employment status and depressive level. Sobel tests showed significant mediation/suppression effects between socio-demographic variables (gender, employment status, and education) and depressive symptoms via vaccine hesitancy. Twenty-eight percent of the total effect of gender on depressive symptoms was mediated by vaccine hesitancy (B = 0.180, *p* = 0.000). About 39% of the effect of employment status on depression was suppressed by vaccine hesitancy (B = −0.254, *p* = 0.000). Lastly, vaccine hesitancy fully mediated the effects of health education on depressive symptoms (B = 0.065, *p* = 0.000).

#### 3.2.3. Factors for Anxiety and Mechanism

Adjusted for background factors, gender (males as reference) (β = 0.05, *p* < 0.05) and employment status (employed as reference) (β = 0.10, *p* < 0.001) was significantly associated with anxiety symptoms. Vaccine hesitancy was also positively associated with anxiety symptoms (β = 0.14, *p* < 0.001) (Figure 3).

As demonstrated in Table 4, Sobel tests showed significant mediation/suppression effects between socio-demographic variables (gender, employment status, and education) and anxiety symptoms. Thirty percent of the total effect of gender on anxiety was mediated by vaccine hesitancy (B = 0.128, *p* = 0.000). About 40% of the effect of employment status on anxiety was suppressed by vaccine hesitancy (B = −0.181, *p* = 0.000). Last, vaccine hesitancy fully mediated the effect of education on anxiety (B = 0.046, *p* = 0.000).

## 4. Discussion

The present study found that 33% of the vaccinated individuals reported some uncertainty before taking the first vaccine shot. The finding suggested that gender, employment status, and vaccine hesitancy were associated with mental health after vaccination. Vaccine hesitancy partially or fully mediated/suppressed the effects between socio-demographics and anxiety/depression. The medication effect indicated that the association between socio-demographics and mental health after vaccination was partially or fully transmitted by vaccine hesitancy. 

In our sample, about one-third of the fully vaccinated participants reported they were hesitant before taking the first vaccine shot. The literature supported our finding by reporting that vaccine hesitancy does not equate to not taking the vaccine [15,36] because vaccine hesitancy is uncertainty, not resistance or refusal. As predicted by the health belief model, people can overcome the hesitancy to receive the vaccine if they perceive more benefits than barriers or perceive high levels of threat from the disease [37]. In this study, people might be hesitant because they are concerned about the safety of the vaccine. However, they would still choose to get vaccinated if they perceived more benefits of taking the vaccine or perceived more threats from the COVID-19 diseases. In addition, the vaccination policy and background in China contribute to hesitant people’s decision to vaccinate. First, different levels of government take strategies to promote vaccination; for instance, communities give incentives, such as produce or money, to encourage vaccination. Furthermore, as the vaccination has been proceeding so rapidly, people might choose to get vaccinated as conformity to society or because of societal pressure. Last, the convenience of taking the vaccine in most areas of China might play as a “cue to action” [37], which also promotes the vaccination rate. As the COVID-19 vaccine has been administered to millions of people in China, this finding implies that a large number of people were hesitant before the vaccination, which suggests a significant gap in health education. 

This study explored the association between socio-demographics and anxiety/depression after vaccination. Females developed more anxious and depressive symptoms after vaccination than males, and the employed developed more anxiety and depressive symptoms than their unemployed counterparts. This finding is consistent with that before the pandemic [7,8,9]. However, few studies have explored the association between demographics and mental distress after vaccination. 

The present study identified the underlying mechanism between socio-demographics and mental health after vaccination. Vaccine hesitancy is directly associated with higher levels of anxiety and depression, and it mediated/suppressed the effects of socio-demographic variables on anxiety and depression. The finding that vaccine hesitancy was positively associated with anxiety and depression after vaccination was supported by recent literature [18,19]. For the mediation effects, first, vaccine hesitancy partially mediated between gender and anxiety/depression (28% and 30% of the total effect, respectively). It indicated that females are more hesitant about vaccination, which relates to higher anxiety/depression levels. Vaccine hesitancy fully mediated the effects of education on anxiety and depression, which suggests that people with a higher education level showed a higher level of anxiety/depression mainly because they were more hesitant about taking the vaccine. Lastly, vaccine hesitancy suppressed the effects of employment status on anxiety and depression, as the employed were more hesitant about vaccination but were less anxious and depressed after vaccination. In summary, vaccine hesitancy transmitted or suppressed partial or total effects of the socio-demographics on mental health. The finding suggests that vaccine hesitancy played a crucial role in people’s mental health after vaccination.

This study found that vaccine hesitancy was significantly associated with gender, employment status, and education. Among our participants, females are more hesitant about receiving the COVID-19 vaccine, which is in line with previous studies in China and abroad [23,24,25,26]. It might be explained that women are more concerned with the side effects and have a higher chance of having medical contraindications, such as during pregnancy and breastfeeding [23,24,25,26]. In addition, a higher education level was associated with higher vaccine hesitancy levels, which is supported by the previous literature [26,27]. This finding might be explained by excessive concerns over vaccine safety in highly educated individuals. We also found that the employed was associated with higher vaccine hesitancy. Worth to note, most of the unemployed (1355 out of 1407) in our sample are students. The previous literature was inconsistent in the association between employment status and vaccine hesitancy [27,28]. Future research is warranted to explore the association between employment status and vaccine hesitancy. In general, we found that females and those in higher socio-economic status (higher education, employed) showed a higher level of vaccine hesitancy, which might be explained by the concerns over the vaccine, which is highly correlated to social and cultural backgrounds. Health education should target these people as the primary population of health promotion programs to eliminate vaccine hesitancy. 

Last but not least, the reported anxiety (2.79%) and depression (3.74%) rates were comparable to the WHO-reported prevalence among the Chinese population [38,39] but lower than the previously reported rates during the COVID-19 pandemic [40,41,42]. There are two plausible explanations for our sample’s lower depression and anxiety rates. First, our study was conducted among people who received the vaccine. These people might have better mental health, as previous studies reported that people with depression and anxiety disorders were less likely to receive the vaccine [43]. In addition, our study was conducted in urban areas in Shanghai, which are well developed in the economy and with good social security. Previous studies reported better mental health status in Shanghai than in other regions of China [44,45]. Additionally, during the investigation, Shanghai was still one of the cities with the best control over the pandemic. 

The finding has important implications for health education. The high prevalence of vaccine hesitancy among vaccinated individuals implies a significant gap in health education to eliminate vaccine hesitancy. The positive association between vaccine hesitancy and anxiety/depression offered insights into the possible mental vulnerability of people holding a hesitant attitude toward vaccination. Public health authorities should be aware of the mental health consequence caused by vaccine hesitancy and pay attention to people’s mental health following massive vaccine campaigns. We found that vaccine hesitancy partially or completely mediated/suppressed the effects of socio-demographic factors on mental distress. It implies that eliminating vaccine hesitancy can decrease the possibilities of mental distress after vaccination. Health promotion programs can target vaccine hesitancy to prevent unfavorable mental health consequences among specific populations. In practice, we can use the health belief model as a framework for health promotion, such as focusing on the benefits of the vaccine, high susceptibility to the virus, and severity of the disease. 

There are a few limitations to mention. The study sample was skewed to a well-educated and younger-aged population, which compromised the sample’s representativeness. As a further limitation, vaccine hesitancy was not assessed by a structured scale but by a question developed after a literature review and discussion with the research team. In addition, recall bias might be an issue since participants need to recall their hesitancy level before their first shot. Furthermore, this study only included individuals who took two doses, omitting those who only took the first dose but not the second, although the latter took a small proportion (2.25%) [1]. This omission might be a source of bias in the association and mediation effects in the results. In addition, this study utilized a correlational approach to assessing the relationships between vaccine hesitancy and mental distress. Thus, causal conclusions cannot be drawn [46]. Prospective cohort studies that track people during the whole vaccination process are recommended in the future. Another limitation is that this study only assessed people’s hesitancy level and mental distress once, and changes in the hesitancy level and mental distress were not captured. Future studies should adopt a longitudinal design to capture changes in vaccine hesitancy and mental distress over time.

## 5. Conclusions

This study examined the association between socio-demographic characteristics and anxiety/depression after vaccination and the mediation effect of vaccine hesitancy underlying the association. We found that vaccine hesitancy mediated/suppressed the effect between socio-demographic characteristics (gender, employment status, and education level) and mental distress. Females and highly educated individuals were significantly more prone to mental distress after being vaccinated, potentially because they were more hesitant about getting vaccinated. This study featured vaccine hesitancy’s potential influence on vaccinated people’s mental health. It is advisory that close attention is paid to mental health following a mass vaccination campaign. Furthermore, communications on the efficacy and safety of the vaccine are essential for eliminating vaccine hesitancy and building public trust in the vaccine. 

## Figures and Tables

**Figure 1 vaccines-10-01697-f001:**
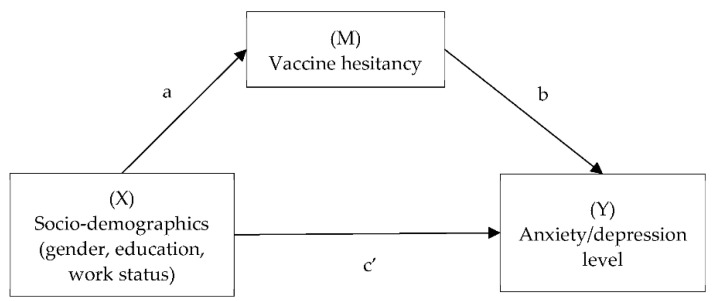
The hypothesized model of mediating effect of vaccine hesitancy on mental distresses. Notes: The mediation model consisted of indirect effect (ab), and direct effect (c′) on outcome. The demographic characteristics were gender, education level, and work status.

**Figure 2 vaccines-10-01697-f002:**
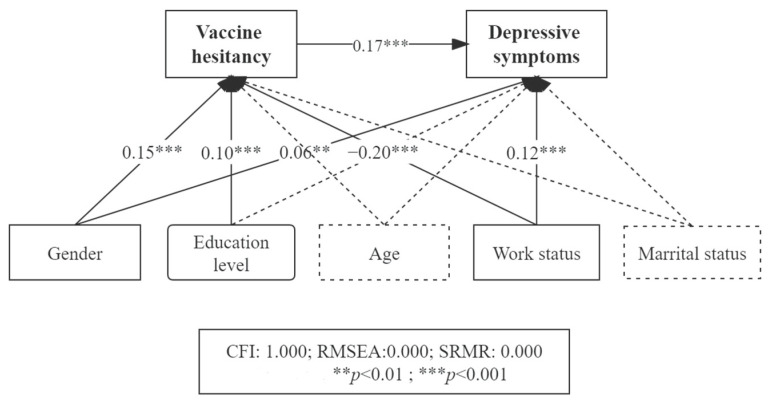
Mediation effect of vaccine hesitancy between demographics and depressive symptoms.

**Figure 3 vaccines-10-01697-f003:**
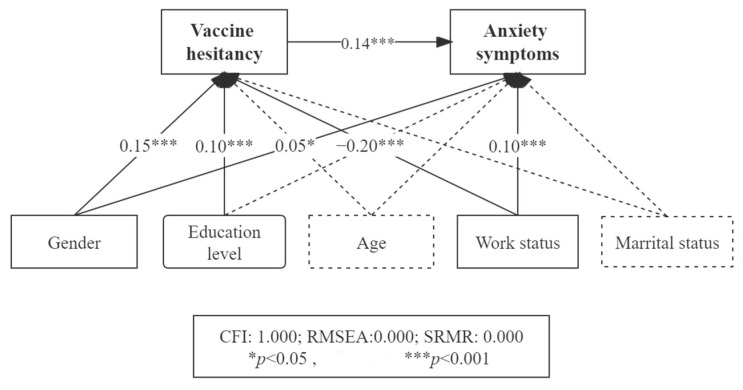
Mediation effect of vaccine hesitancy between demographics and anxiety symptoms.

**Table 1 vaccines-10-01697-t001:** Sample characteristics of participants (*N* = 2112).

Characteristics	Number (*n*), %	Mean (SD)
**Age (years)**		
18–25	63.26%	
26–40	26.70%	
>40	10.04%	
**Gender**		
Female	53.60%	
Male	46.40%	
**Education level**		
<High school	10.75%	
≥High school	89.25%	
**Employment status**		
Employed	31.30%	
Unemployed (including students)	68.70%	
**Marital status**		
Unmarried	78.08%	
Married	20.93%	
Divorced or widowed	0.99%	
**Areas of living**		
Urban	91.05%	
Non-urban	8.95%	
**Hesitancy**		
Not hesitant at all	30.87%	
Merely hesitant	33.76%	
Neutral	2.08%	
Somewhat hesitant	31.82%	
Very much	1.47%	
**Depression**		
No depression (0–4)	84.52%	
Mild depression (5–9)	11.74%	
Moderate or major (≥10)	3.74%	
**Depression** (continuous)		1.95 (3.65)
**Anxiety**		
No anxiety (0–4)	86.93%	
Mild anxiety (5–9)	10.27%	
Moderate or major anxiety (≥10)	2.79%	
**Anxiety** (continuous)		1.51 (3.04)

**Table 2 vaccines-10-01697-t002:** Estimated correlations between variables of interest (*N* = 2112).

	Age	Gender	Education	Employment Status	Marital Status	Vaccine Hesitancy	Anxiety	Depression
Age	1.00							
Gender	0.05 *	1.00						
Education	0.07 *	0.10 *	1.00					
Employment status	−0.64 *	−0.04	−0.00	1.00				
Marital status	0.76 *	0.05 *	−0.10 *	−0.53 *	1.00			
Vaccine hesitancy	0.17 *	0.16 *	0.11 *	−0.22 *	0.14 *	1.00		
Anxiety	−0.11 *	0.12 *	0.09 *	0.12 *	−0.12 *	0.15 *	1.00	
Depression	−0.13 *	0.12 *	0.09 *	0.12 *	−0.14 *	0.15 *	0.74 *	1.00

* Spearman’s correlation coefficient.

**Table 3 vaccines-10-01697-t003:** The mediation effect of vaccine hesitancy between demographics and depression.

	X→M	M→Y	X→Y	Sobel Test	RIT (Indirect Effect/Total Effect)	RID (Indirect Effect/Direct Effect)
Gender	B = 0.374, *p* = 0.000	B = 0.481, *p* = 0.000	B = 0.472, *p* = 0.003	B = 0.180, *p* = 0.000	0.276	0.381
Employment status	B = −0.529, *p* = 0.000	B = 0.481, *p* = 0.000	B = 0.914, *p* = 0.000	B = −0.254, *p* = 0.000	0.385	0.278
Education level	B = 0.135, *p* = 0.000	B = 0.481, *p* = 0.000	B = 0.017, *p* = 0.842	B = 0.065, *p* = 0.000	0.788	3.715

**Table 4 vaccines-10-01697-t004:** The mediation effect of vaccine hesitancy between demographics and anxiety.

	X→M	M→Y	X→Y	Sobel Test	RIT (Indirect Effect/Total Effect)	RID (Indirect Effect/Direct Effect)
Gender	B = 0.374, *p* = 0.000	B = 0.342, *p* = 0.000	B = 0.299, *p* = 0.023	B = 0.128, *p* = 0.000	0.299	0.427
Employment status	B = −0.529, *p* = 0.000	B = 0.342, *p* = 0.000	B = 0.636, *p* = 0.023	B = −0.181, *p* = 0.000	0.397	0.284
Education level	B = 0.135, *p* = 0.000	B = 0.342, *p* = 0.000	B = 0.073, *p* = 0.321	B = 0.046, *p* = 0.000	0.386	0.629

## Data Availability

The data that support the findings of this study are available from the School of Medicine, Tongji University, but restrictions apply to the availability of these data, which were used under license for the current study. Data are available from the authors upon reasonable request and with permission of Shanghai Pudong New Area Mental Health Center, School of Medicine, Tongji University.

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
