# Peer review of "The Association between Socio-Demographics and Mental Distress Following COVID-19 Vaccination—Mediation of Vaccine Hesitancy"

_vaccines, 2022, doi:10.3390/vaccines10101697_

Round 1
Reviewer 1 Report
Estimated Authors of the study "The association between COVID-19 vaccine hesitancy and negative psychiatric outcomes among Chinese vaccinated individuals—a path analysis study", I've read with your article with interest, but also with great pleasure.
From my point of view, this is a very interesting paper, well written and detailed, that will fill some significant knowledge gaps on our understanding of Vaccine Hesitancy. In this study, Zhang et al. have been able to clearly and steadily associate COVID-19 VH with a series of psychiatric issue, and more precisely that individuals with a higher level of hesitancy before receiving the vaccine developed more anxiety and depressive symptoms three weeks later.
Not only I've no specific requests of recommendations, but I'm recommending the acceptance of this paper as it is, with high priority.
Author Response
Thank you very much for the encouraging words.
Reviewer 2 Report
interesting paper. I think that Authors should indicate esclusion criteria. I think that incentives as compensation for time spent could constitute a sufficient reason for filling a questionnaire in a total random way... and I think that it can be difficult to discriminate between people really interested in participating in the study...
Table 1 may contain an error (highlighted)

Author Response
Thank you for the suggestions.
- Exclusion criteria are added in the text (lines 94-95).
“Participants younger than 18 and those who received the first shot within the past 21 days were excluded from the study.”
- We used incentives to attract participants, but we also adopted various ways to control the quality of the survey.
First, we chose a small package of sanitizer wipes as an incentive because it can attract people to participate but is not as expensive as attracting people without any interest in the survey.
In addition, we took solutions to control the quality of the investigation. On each investigation, we staffed questionnaire interviewers to help with the answering process.
Interviewers checked the completed questionnaire of each participant before distributing the incentives. Last, we also set answering time length as a way of scrutinizing the poor-quality answers. Responses finished within 120 seconds were excluded from the analysis, based on the length of the questionnaire.
Reviewer 3 Report
Known in the field based on previous literatures:
1. Coronavirus disease 2019 (COVID-19), the ongoing pandemic, is an infectious disease caused by the most recently discovered coronavirus- severe acute respiratory syndrome coronavirus 2 (SARS-CoV-2).
2 Infected people reported – mild symptoms to severe illness and COVID-19 affects different people in different ways. Symptoms of COVID-19 are variable, and mainly include, cough, breathing difficulties, headache, fever, and some time loss of smell and taste.
3 COVID-19 vaccination program has been applied globally since December 2020 to counteract the pandemic, but various side effects are also reported.
In this article authors reported following findings:
In this article, authors studied the psychiatric impact of vaccine hesitancy among the vaccinated population. Authors reported following findings-
1. The individuals with a higher level of hesitancy before receiving the vaccine developed more anxiety and depression after vaccination.
2. There are many factors which are associated with vaccine hesitancy like gender, education, and employment.
3. Authors suggested authority or Government should pay attention about vaccine hesitancy, and education and awareness could help to overcome anxiety and depression against vaccine.
Although, authors nicely mentioned many facts related to COVID-19 vaccine hesitancy and negative psychiatric outcome but there is nothing novel, and the only difference is number of samples size and site of study. The following suggestions if incorporated could help in the better understanding of the significance of the work and implications.
Minor Concerns:
1. The various social and personal causes (unemployment and personal mental status) also lead to anxiety and depression how authors differentiated with vaccine hesitancy?
2. Please explain why and how more educated females were associated with vaccine hesitancy than male? Is it related with sex hormones?
3. Authors could also add a table having different side effect of COVID-19 among male vs female and educated vs uneducated population.
